# Study protocol for the antidepressant advisor (ADeSS): a decision support system for antidepressant treatment for depression in UK primary care: a feasibility study

Phillippa Harrison ![ORCID],[1] Ewan Carr,[2] Kimberley Goldsmith,[2] Allan H Young,[1] Mark Ashworth ![ORCID],[3] Diede Fennema,[1] Barbara Barrett,[4] Roland Zahn[1]

**Correspondence to**
Dr Roland Zahn;
Roland.zahn@kcl.ac.uk

## ABSTRACT

**Introduction** The Antidepressant Advisor Study is a feasibility trial of a computerised decision-support tool which uses an algorithm to provide antidepressant treatment guidance for general practitioners (GPs) in the UK primary care service. The tool is the first in the UK to implement national guidelines on antidepressant treatment guidance into a computerised decision-support tool.

**Methods and analysis** The study is a parallel group, cluster-randomised controlled feasibility trial where participants are blind to treatment allocation. GPs were assigned to two treatment arms: (1) treatment-as-usual (TAU) and (2) computerised decision-support tool to assist with antidepressant choices. The study will assess recruitment and lost to follow-up rates, GP satisfaction with the tool and impact on health service use. A meaningful long-term roll-out unit cost will be calculated for the tool, and service use data will be collected at baseline and follow-up to inform a full economic evaluation of a future trial.

**Ethics and dissemination** The study has received National Health Service ethical approval from the London—Camberwell St Giles Research Ethics Committee (ref: 17/LO/2074). The trial was pre-registered in the Clinical Trials.gov registry. The results of the study will be published in a pre-publication archive within 1 year of completion of the last follow-up assessment.

**Trial registration number** NCT03628027.

## INTRODUCTION
### Background and rationale

Depression is a leading cause of disability worldwide[1] with approximately 2 million adults in the UK experiencing an episode each week.[2] Population-based studies show that only 50% of people with major depressive disorder (MDD) are treated[3 4] and more than half of these receive inadequate treatment.[4] Furthermore, less than one-third of patients fully recover after treatment with a standard antidepressant, such as citalopram,[5] and only 40% of patients recover after

<div class="box">

**Strengths and limitations of this study**

▶ A strength of the study is integration of a decision-support tool into an existing UK primary care computer system to reduce the burden on general practitioners (GPs) of adopting a new treatment system.

▶ A further strength is that the study allows patients to be treated by their regular GP practice which reduces the burden on patients.

▶ A limitation of the study is the potential for selection bias towards those with less severe depression who are well enough to attend study assessments.

</div>

accessing the National Health Service (NHS) Improving Access to Psychological Therapies programme.[6] The UK National Institute for Health and Care Excellence (NICE) recommends three first-line antidepressants with similar mechanism of action (fluoxetine, sertraline and citalopram).[7] Second-line and third-line treatments are recommended by NICE, but, due to a lack of sequenced guidance, it is unclear whether and when general practitioners (GPs) should prescribe them.[7] Overall, unlike for the management of other long-term conditions, there is no structured disease management for antidepressant prescribing in UK primary care[8 9] and patients' access to second-line and third-line antidepressants is variable. National prescription data show frequent and rising use of certain second-line antidepressants such as venlafaxine and mirtazapine without clear decision strategies.[10] A lack of structure in treatment decision making and poor implementation of existing guidelines could exacerbate the suffering caused by insufficiently treated depression.[4] Enhanced depression

**BMJ**

care has been called for and entails tailoring treatment to measured outcomes.[9]

One way to provide structured treatment guidelines is through algorithms which incorporate various patient characteristics and allocate treatments most likely to be effective. To the authors' knowledge, there is no scientifically evaluated and pragmatic antidepressant decision-support tool in UK primary care. This is based on a literature search conducted in September 2019 in the International Standard Randomised Controlled Trial Number (ISRCTN), Eudra CT, Clinical Trials.gov, PubMed and Web of Knowledge databases using the key words: ("depression or depressive") AND "decision" AND ("computer or algorithm") AND "antidepressant". The search returned four relevant studies which employed a computer-based algorithm to help make decisions about antidepressant treatments.[11–14]

Out of three completed studies, two found evidence for the effectiveness of algorithm support compared with treatment-as-usual. Adli et al[13] found shorter time to remission and fewer medication switches using an algorithm compared with computerised support and treatment-as-usual for depression treatment. Kurian et al[11] assigned private GPs and their MDD patients to active computerised decision support for antidepressants based on the 1999 Texas Medication Algorithm Project[15] and treatment-as-usual. The active group showed a 41% response rate to treatment versus 26% in the treatment-as-usual group. However, Rollman et al[12] reported no difference in depressive symptoms after 6 months between active decision support, non-specific prompting that depression needs treatment and treatment-as-usual. The ongoing Assurex Health Inc Study[14] uses a genetic testing decision-support tool to predict differential treatment response to treatments for depression.

Overall, all of the above studies were conducted outside of the UK, often in private healthcare settings which are difficult to compare with the UK NHS. Several of the support tools did not provide specific treatment sequences, only broad recommendations to change medication. Furthermore, some of the studies could not translate recommendations into algorithms to be used by GPs so that researchers had to manually provide GPs with advice reports. This creates a complicated treatment process and it seems unlikely that the support tools could provide ongoing recommendations to prescribers after initial treatment recommendations. Moreover, Kurian et al[11] and Rollman et al[12] employed algorithms that are now outdated and lacked integration of recent evidence on superiority of specific novel antidepressants in terms of tolerability and efficacy. All of the tools lacked integration into existing electronic healthcare record systems that GPs are familiar with. NICE guidelines provide useful MDD treatment advice for GPs but a scarcity of studies on complex interventions using sequenced treatment algorithms has meant that the guidelines are unclear about how many switches between individual antidepressants are recommended

before seeking specialist advice on other strategies such as, augmentation. The seminal STAR*D Study investigating such a sequenced treatment algorithm has used augmentation strategies that according to NICE should not be carried out by GPs.[16]

Therefore, the aim of the current study is to assess the feasibility of a future definitive randomised controlled trial to test the efficacy and cost effectiveness of a decision tool incorporating an algorithm to advise GPs on antidepressant prescribing for patients who have not responded to first-line treatments. In contrast to earlier work, the treatment algorithm is based on NICE guidelines including the latest health technology appraisals. If successful, the present study will therefore enable the first investigation into whether a sequenced treatment algorithm that relies on well-tolerated single antidepressants is effective and cost effective. The authors have gained support from one of the leading UK providers of primary care electronic record systems (EMIS group) who have implemented the algorithm into their software. The tool will be designed to be easy to use and to save time for GPs who are often limited to 10 min appointments.

## STUDY OBJECTIVES
1. To assess the feasibility of a future confirmatory trial investigating the first computerised decision-support tool for antidepressant treatment in UK primary care by:
   a. estimating lost to follow-up rates.
   b. estimating GP adherence to the algorithm and patient adherence to prescribed medications.
   c. determining the number of GP practices willing to recruit patients for the study (determined by the Clinical Research Network (CRN) who will approach all practices in the participating Clinical Comissioning Groups (CCG)).
   d. estimating participant recruitment rates per GP.
   e. estimating GP satisfaction with the decision tool.
2. To provide initial SD estimates and intra-class correlation coefficients for computing effect size estimates for a larger confirmatory trial.
3. To collect health economic estimates of the roll-out cost of the intervention and changes in service use associated with the tool, including psychiatric referrals to mental health teams and/or the study psychiatrist.

## METHODS
### Trial design
The study design is a single-blinded cluster-randomised controlled clinical trial, where the GP practice is the cluster/randomisation unit. Considering complex intervention guidance,[17] the study is designed as a feasibility study of the decision-support tool to provide estimates of the unknown variables needed to plan a subsequent larger trial.

## Study setting

The study is based in UK primary care, recruiting GPs and patients from South London primary care practices. Patients will be assessed at the Institute of Psychiatry, Psychology and Neuroscience in South London and treated at their usual GP practice. The study start date was August 2018 and the planned end date is July 2021.

## Participants

### Eligibility criteria for patients

#### Inclusion criteria

Age ≥18; at least moderately severe major depressive syndrome on Patient Health Questionnaire (PHQ-9; score ≥15)[18]; no plans to change GP practice; able to complete self-report scales orally or in writing; no previous prescription of mirtazapine or vortioxetine and early treatment resistance. The latter is defined as (1) current or recent prescription (in previous 2 months) of any of the following antidepressants: citalopram, fluoxetine, sertraline, escitalopram, paroxetine, venlafaxine or duloxetine and (2) previous prescription of at least one other antidepressant from the same list.

#### Exclusion criteria

Inability to consent to the study; unstable medical condition; currently being treated by mental health specialist; high suicide risk (Mini International Neuropsychiatric Interview (MINI) suicidality screen)[19]; past diagnosis of schizophrenia or schizo-affective disorder; current psychotic symptoms (three clinical screening questions to exclude schizophreniform disorders)[20 21]; bipolar disorder (WHO Composite International Diagnostic Interview (CIDI) screening scale for bipolar disorder)[22]; currently at risk of being violent; drug or alcohol abuse over the last 6 months (Primary Care Evaluation of Mental Disorders (PRIME-MD)[23], modified to screen for drug abuse); breastfeeding/within 6 months of giving birth and both escitalopram and sertraline have already been prescribed.

### Participant timeline

Potential participants will be identified using an EMIS eligibility tool to search participating GP practice medical records for patients meeting inclusion criteria (see online supplementary file 1). Interested participants will complete a pre-screening assessment, followed by a screening assessment including a clinical evaluation to determine full eligibility. The results of the screening assessment will be shared with GPs and patients. Inclusion decisions will not be made by GPs to help mitigate against selection biases and will be decided by the research team after the screening assessment. After being enrolled, patients receive access to a secure mobile phone app developed in the study. Eligible patients will undergo treatment for depression over 14 weeks with their GPs. In both arms, patients will be seen by their GP who will review and monitor treatment. Patients' participation will last for approximately 15–18 weeks after their screening assessment, at which point their follow-up assessment will be conducted. See table 1 for participant timeline.

### Interventions

Participants meeting the eligibility criteria will participate in the intervention arm to which their GP practice is randomised:

Arm 1: Treatment-as-usual (TAU). Treatment as GPs would usually deliver with no decision support and no

**Table 1** Participant timeline

| | Participants identified by EMIS search tool | Consent for contact | Initial pre-screening (electronic, letter or phone) | Pre-trial screening assessment in person | Treatment sessions arranged by GP as soon as possible after pre-trial assessment with no fixed number over 14 weeks | Post-trial assessment in person after 15–18 weeks since pre-trial assessment |
|---|---|---|---|---|---|---|
| Participants contacted about study | X | | | | | |
| Reply by letter slip, phone, text or email | | X | | | | |
| Oral/electronic or written pre-screening informed consent | | | X | | | |
| Introduction to the study | | | X | | | |
| Assessment of eligibility | | | X | X | | |
| Written informed consent | | | | X | | |
| Clinical assessment and neuro-psychological tests | | | | X | | X |
| Trial intervention delivered by GPs | | | | | X | |
| Mobile app/phone weekly assessment | | | | | X | |

GP, general practitioner.

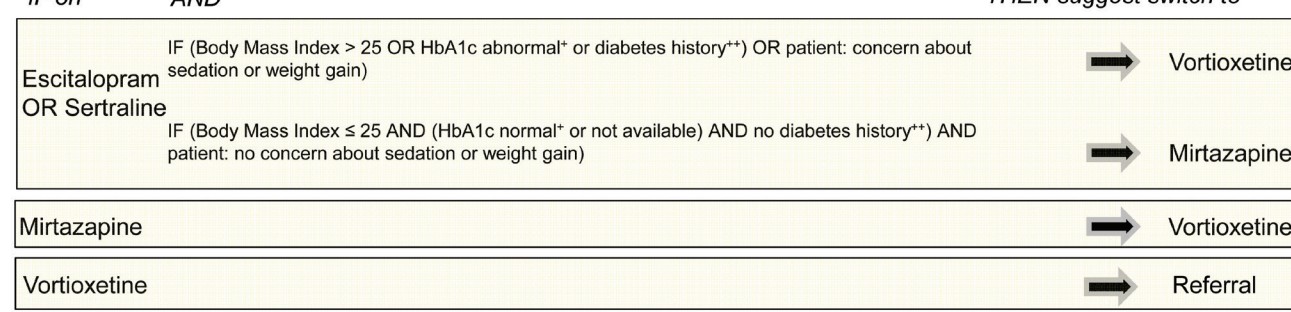

Step 1) a) Algorithm checks whether a current antidepressant from eligibility list has been prescribed at sufficient dose[1] for ≥4 weeks
*IF a) is FALSE THEN ASK GP whether this antidepressant has been*
b) Stopped because not tolerating side effects? (If "no" to b), THEN display message: "If other reason for stopping, please review treatment"
*IF a) prescription has been made for less than 4 weeks* THEN END advisor module *and Display message to repeat PHQ-9*
*after 4 weeks from initial prescription*
*IF "yes" to a) OR b)* *THEN suggest switch to*
*AND*

| | | |
|---|---|---|
| IF "no" to b) | IF (no escitalopram previously prescribed AND sertraline previously prescribed) | ➡ Escitalopram |
| | IF (no sertraline previously prescribed) | ➡ Sertraline |
| IF "yes" to b) | IF (Body Mass Index ≤ 25 AND (HbA1c normal+ or not available) AND no diabetes history++) | ➡ Mirtazapine |
| | IF(Body Mass Index > 25 OR HbA1c abnormal+ OR diabetes history++) | ➡ Vortioxetine |

Step 2) IF ≥3 weeks of prescription with initial dose[2] AND (insufficient PHQ-9 drop from baseline* AND >9) AND tolerated, THEN suggest increase to high dose[3]

Step 3) IF (≥3 weeks of prescription with high dose[3] OR not tolerated) AND (insufficient PHQ-9 drop from baseline* AND >9):

*IF on* *AND* *THEN suggest switch to*

| | | |
|---|---|---|
| Escitalopram OR Sertraline | IF (Body Mass Index > 25 OR HbA1c abnormal+ or diabetes history++) OR patient: concern about sedation or weight gain) | ➡ Vortioxetine |
| | IF (Body Mass Index ≤ 25 AND (HbA1c normal+ or not available) AND no diabetes history++) AND patient: no concern about sedation or weight gain) | ➡ Mirtazapine |
| Mirtazapine | | ➡ Vortioxetine |
| Vortioxetine | | ➡ Referral |

For each Step, call EMIS prescribing tool and IF high severity warning THEN advise: "Skip to next step unless you think warning is based on irrelevant information"
For each Step, call EMIS prescribing tool and enter cross-tapering scheme into prescribing information (see further cross-tapering definitions)
Before each recommended step ask GP whether drug tolerated, if not, then skip to next drug recommendation (optionally EMIS displays side effects collected via mobile app)
Zahn et al., Primary Care Antidepressant Advisor Algorithm software requirements, 3.1.2018

**Figure 1** Decision chart for decision-support tool. GP, general practitioner; PHQ-9, Patient Health Questionnaire.

constraints (these GPs will not be aware of the rules of the treatment algorithm used in the other group nor will they be able to access it);

Arm 2: Decision-support tool. Treatment using the decision-support tool to assist GPs with antidepressant prescriptions, prompting GPs to review patients' medications and change them if ineffective. The algorithm and technical requirements of the tool are described below.

An EMIS decision support module was developed which provides suggestions for antidepressant prescribing after highlighting the importance of non-pharmacological strategies such as psychological therapy or exercise. The treatment recommendations are based on NICE guidelines, British Association for Psychopharmacology guidelines[24] and Maudsley prescribing guidelines.[25] The algorithm is designed to incorporate patient preferences, ensure maximum prescribing safety and offer guidance on antidepressant monotherapy (avoiding combinations). NICE guidelines state that GPs should not normally initiate combinations without consulting a psychiatrist first.[7] See figure 1 for stepped recommendations provided in the tool; for further detail see online supplementary file 1.

The module is implemented only for those GP practices taking part in the study and only for participating patients at those practices. GPs will use the EMIS module during patient appointments by entering patients' Maudsley Modified Patient Health Questionnaire (MM-PHQ-9)[26] scores of their depressive symptoms over the last week from the study mobile app. The tool will offer treatment advice on whether a patient's antidepressant should be maintained or changed and what it should be changed to. The advisor tool is used for every depression review appointment over the 14-week study duration, with no minimum number of appointments.

GPs in both conditions will receive training by the study Principal Investigator (PI) on study procedures including adverse event (AE) reporting and incorporating patients' mobile app data. Training will be tailored to treatment arm so that GPs in the decision-support tool arm will be trained in how to use the EMIS module to guide treatment decisions whereas GPs in the TAU arm will be advised to use their clinical judgement as usual for treatment.

### Adherence

Adherence to the prescribed drugs in both arms will be monitored using the available tools on EMIS as well as the mobile app. Patients will be withdrawn from the antidepressant early in the event of a serious or medically important AE considered by the GP to be related to the intervention, hypomania or a significant deterioration of symptoms requiring referral to secondary care. GP adherence to the algorithm will be measured after GPs have

completed their participation in the study with a self-report questionnaire and information from EMIS.

## Outcomes
### Feasibility outcome measures
1. Lost to follow-up rates.
2. GP adherence to the algorithm for each completed patient rated by a trial clinician (0–3 for none, less than 50%, 50% or more or 100% of recommended steps implemented).
3. Average patient adherence to prescribed medications based on EMIS electronic prescribing records.
4. AE and serious adverse event (SAE) rates (grade and relationship to intervention).
5. Patient adherence to GP attendance measured by % of attended GP visits out of scheduled visits on EMIS over treatment period.
6. Recruitment rates.
7. Average GP satisfaction with decision-support tool (intervention arm; after GP completion of study).
8. Diagnostic Statistical Manual (DSM)-IV Social and Occupational Functioning Assessment Scale of psychosocial functioning (on final visit, while modelling baseline score).[27]
9. Maudsley Visual Analogue Mood Scale (on final visit, while modelling baseline score).

This section describes outcomes that will be measured in this feasibility trial to provide estimates (eg, effect size, SD) needed for sample size calculations for a future confirmatory trial. These outcomes will not be used for inferential comparisons between trial arms.

### Primary clinical outcomes for a future confirmatory trial
10. Self-rated Quick Inventory of Depressive Symptomatology Sum Score (QIDS-SR16)[28] at final visit, adjusting for baseline score.

### Secondary clinical outcome measures for a future confirmatory trial
11. Depressive symptoms assessed by the Montgomery–Asberg Depression Rating Scale at follow-up assessment, adjusting for baseline score.[29]
12. Clinical Global Impression Scale; change between baseline and follow-up assessment (final visit).[30]
13. Generalised Anxiety Disorder-7 Scale at follow-up assessment, adjusting for baseline score.[23]
14. Body mass index at follow-up assessment adjusting for baseline score.

### Exploratory clinical outcome measures (self-report via mobile app)
15. Average score for medication side effects on Frequency, Intensity and Burden of Side Effects Ratings (FIBSER).[31]
16. Average % of adherence to prescribed antidepressant medication.
17. Average MM-PHQ-9 scores in last 2 weeks (at follow-up, while modelling first 2 weeks as baseline average).[26]

### Health economic measures
18. Service use as determined on EMIS including psychiatric referrals and referrals to study psychiatrist, as well as time to psychiatric referral; also primary care consultation rates.
19. Service use; self-reported using a modified version of the Adult Service Use Schedule.[32]
20. Quality of life using the EQ-5D-3L[33]—the standard measure recommended by NICE for cost-effectiveness analyses.

## Sample size
Each practice will be asked to enrol approximately 8–11 participants. The study aims to enrol 86 participants assuming the same lost to follow-up rate as in Kurian *et al*'s study[11] (18%), giving a final sample size of 70 (35 in each group as recommended[34]). This will enable estimation of the lost to follow-up rate within a 95% CI of ±8%.[35] Kurian *et al*[11] did not provide effect size estimates so the present study has been designed to provide means and SDs, as well as CIs for measures of change on the primary outcome measure (QIDS-SR16)[28] as recommended for feasibility trials.[34]

## Recruitment of GP practices
The study aims to recruit 8–20 GP practices. Only one GP per practice will be recruited to the study to avoid communication about treatment allocation between GPs. In the event of a GP leaving a participating practice, an alternative GP from the same practice will replace them.

## Enrolment of participants
Patients will be enrolled after GPs are randomised. An EMIS eligibility tool developed for the study will identify potential participants from each practice meeting the inclusion/exclusion criteria (see online supplementary file 1 for further details). Practice/CRN staff will run the EMIS search and send eligible patients a study advert and request for consent for contact via post.

## Assignment of interventions
### Allocation
#### Sequence generation
The randomisation service will be provided by the King's Clinical Trials Unit (KCTU) in accordance with a standard operating procedure and held on a secure server. GP practices will be randomised in pairs with a 1:1 allocation to either intervention or TAU using block randomisation with random permuted blocks of block size 2. This will help maintain allocation concealment while ensuring a similar number of practices are allocated to each arm.

#### Concealment mechanism
KCTU will send the randomisation outcome to unblinded researchers only to ensure researchers conducting assessments remain blind. The randomisation details will be kept on a password-protected network drive only accessible by the PI.

## BLINDING

Patients, researchers assessing observer-rated outcomes and the senior statistician will be blind to treatment allocation. All other members of the study team, including the junior statistician, and GPs will be unblinded. Unblinding incidents will be recorded in the trial master file and reported to the Trial Steering Committee (TSC; see the Data monitoring section). Unblinded assessments will be retained in the study. If a researcher becomes unblinded prior to completing assessments, an alternative researcher will conduct future assessments with patients from that practice in order to conduct unbiased assessments. There is no requirement for an emergency unblinding procedure, as the PI and GPs are not blinded.

## DATA COLLECTION METHODS

The pre-screening assessment is conducted online using a survey software (alternatively over the telephone or via postal survey) and includes the PRIME-MD (self-report)[23] to assess depressive symptoms and alcohol abuse and modified to screen for drug abuse, the CIDI (self-report) for bipolar disorder,[22] questions to exclude schizophreniform disorders and psychotic depression,[20 21] and pregnancy in women. Eligible participants are invited to a face-to-face screening assessment which includes depression history, treatment history, medical history, MINI suicidality screen,[19] DSM-5 Structured Clinical Diagnostic Interview,[27] Psychiatric Family History Screen,[36] *Arbeitsgemeinschaft für Methodik und Dokumentation in der Psychiatrie* (AMDP) Psychopathology Interview questions of depression items,[37 38] Addenbrooke's Cognitive Examination-III in patients over 50 to detect early Alzheimer's disease,[39] Longitudinal Interval Follow-up Evaluation (LIFE)[40] and the Young Mania Rating Scale (YMRS).[41] Participants' medical records will be checked to confirm medical details. Participants will use the mobile app for the 14-week study duration to enter weekly MM-PHQ-9 ratings,[26] hypomanic symptoms,[22] FIBSER medication side effects,[31] medication changes and self-blame-related action tendencies such as to what extent participants would feel like hiding or creating a distance from themselves (using two questions developed in Dr Zahn's research lab). A daily question will ask about medication adherence. Alternatively, participants can opt to provide this data at weekly intervals via post or phone.

At the follow-up assessment, the study outcome measures and YMRS[41] will be repeated. The LIFE[40] will be used to determine remission and its psychiatric status rating scale used as a comparison to baseline. All assessments will be conducted by trained researchers. The research team will be trained and closely supervised, establishing sufficient inter-rater reliability on semi-structured interviews with the PI before conducting assessments independently.[21]

## PATIENT AND PUBLIC INVOLVEMENT

The study is supported by a service user advisory group which provides input to the study. This group meets on a regular basis and provides insight into the study design, information disseminated to patients and the burden of trial participation from the patient's perspective. At the end of the study, the service user advisory group will comment on the findings and contribute to the dissemination plan.

## STATISTICAL METHODS
### Outcomes

Categorical outcomes (eg, loss to follow-up) will be described using frequencies and proportions. The QIDS-SR16 and other continuous outcomes will be summarised at baseline and follow-up using means and SDs. The GP practice intra-class correlation will be calculated for the outcome variable using one-way random effects analysis of covariance (adjusted for baseline). A preliminary analysis of the difference between the groups, as far as possible using the intention-to-treat principle, is planned. This analysis will be identified as preliminary and underpowered when published, and no p values will be provided. Continuous outcomes measured at baseline and follow-up only, such as the QIDS-SR16, will be analysed using linear regression with robust SE calculation to account for clustering within GP practice. Continuous outcomes measured weekly will be analysed using mixed linear regression models with a random intercept for GP practice to account for clustering. Both sets of models will include treatment arm as a covariate and will adjust for baseline measure of the outcomes where appropriate. Any missing baseline data will be imputed using mean imputation.[42] Missing data in weekly outcomes will be accounted for under the missing at random assumption by using the maximum likelihood algorithm to fit the mixed models. We may consider multiple imputation for outcomes measured only at follow-up if post-randomisation variables can be quantified and are related to having missing outcomes. AEs (see the Harms section) will be summarised separately as AEs and SAEs, by intervention group, as number of events and number of people experiencing events.

### Data monitoring

The TSC will meet bi-annually to oversee study progress and comprises the study PI, research associate, research assistants, statisticians, collaborators and service users. There are no formal criteria for terminating the trial. Informal decisions about early trial termination will be made by the TSC and sponsor should they have major concerns about safety or conduct of the trial. This would be considered in the event that extensive recurrent serious or medically important AEs are observed, particularly if these are Suspected Unexpected Serious Adverse Reactions.

## Harms

GPs participating in the study, especially in the intervention arm, are likely to change medications more often and prescribe medications which are less frequently used in primary care. All medications recommended by the decision-support tool are in accordance with NICE guidance and were carefully selected for the study based on likelihood of positive effects and side effects (see online supplementary file 1). Kurian *et al*[11] found their support tool to be beneficial for depression treatment, although some patients felt worse as a result of changing medications—for example, having more suicidal thoughts. Such side effects will be monitored by GPs and discussed with patients.

AEs/SAEs are defined in online supplementary file 1. AEs/SAEs will be recorded from the time the subject provides informed consent at the baseline assessment until their follow-up. GPs will record all AEs/SAEs on EMIS and they will be regularly collected from GPs by the research team and assessed. AEs/SAEs reported to the PI will be recorded on AE forms as part of the case report form using an ID number to identify the patient. AEs/SAEs will be reported bi-annually to the TSC and an unblinded sub-committee, assuming the functions of the data monitoring committee, will review the information separated by arm.

### Ethics and dissemination

#### Research ethics approval

The study has received NHS ethical approval from the London—Camberwell St Giles Research Ethics Committee (ref: 17/LO/2074).

#### Protocol amendments

Modifications to the protocol will be conveyed to the TSC at the planning stage for members to provide input, then again to inform the committee of the ethics committee decision.

#### Consent

Participants will provide informed consent to take part in the study and are free to withdraw at any time with no reason given (see online supplementary files 1 and 2 for further detail).

#### Dissemination

The trial has been pre-registered on ClinicalTrials.gov. See online supplementary file 1 for further detail on dissemination.

### Health economics

The study will not include a formal economic evaluation, instead the focus is on developing the tools needed to support a future trial. First, a meaningful unit cost will be calculated for our tool in collaboration with EMIS. The unit cost will consider the long-term roll-out cost of the intervention and investigate how to allocate this to an individual patient in order to carry out a future cost-effectiveness analysis. Second, the methods for the collection of service use data will be optimised, which will include a comparison of data collected from EMIS and data collected from patients via self-report.

## LIMITATIONS

One limitation of the algorithm is that it does not take pharmacogenomic evidence into account which could be an important refinement in the future, should pharmacogenomic testing become implemented into NHS routine care. For example, both CYP2D6 and CYP2C19 gene polymorphisms affect the liver metabolism of sertraline, escitalopram and citalopram, as well as paroxetine and fluvoxamine.[43] Therefore one could argue if someone has not responded to or not tolerated citalopram or paroxetine, this carries a higher risk of non-tolerance or non-response of escitalopram and sertraline. Pharmacogenomic effects of liver enzymes, however, are only one of many factors which determine response and tolerance.[44] From a pragmatic perspective, it is therefore worth trying escitalopram or sertraline as the first step in the algorithm which is supported by network-meta-analytical evidence.[45] CYP2D6 also plays a role in metabolising vortioxetine and mirtazapine, but pharmacogenomic guidelines have not been developed for these to our knowledge.

**Author affiliations**
[1]Centre for Affective Disorders, Department of Psychological Medicine, Institute of Psychiatry, Psychology & Neuroscience, King's College London, London, UK
[2]Department of Biostatistics and Health Informatics, Institute of Psychiatry, Psychology & Neuroscience, King's College London, London, UK
[3]Department of Population Health Sciences, King's College London, London, UK
[4]Department of Health Services & Population Research, Institute of Psychiatry, Psychology & Neuroscience, King's College London, London, UK

**Acknowledgements** This paper represents independent research funded by the National Institute for Health Research (NIHR) research for patient benefit scheme. This paper represents independent research part funded (KG, EC) by the National Institute for Health Research (NIHR) Biomedical Research Centre at South London and Maudsley NHS Foundation Trust and King's College London. The views expressed are those of the authors and not necessarily those of the NHS, the NIHR or the Department of Health and Social Care. DF's PhD is funded by the Medical Research Council Doctoral Training Partnership (project reference: 2064430). The authors would like to acknowledge the support provided to the study by the South London Clinical Research Network and sponsorship by Lambeth CCG. Additionally, the authors wish to thank the following members of the Trial Steering Committee who have dedicated their time and provided valuable input to the study: Dr Daniel Dietch, Dr Victoria Cornelius, Evelyn London and Dr Sarah Markham. We are also grateful to EMIS PLC with whom we have designed the software implementation of our Antidepressant Advisor decision-support tool and to Alloc Modulo LTD with whom we have developed the accompanying MooDoC mobile app.

**Contributors** PH wrote the manuscript. EC, KG, AHY, MA, DF, BB and RZ all commented significantly on drafts of the manuscript. EC and KG provided the statistical analysis plan. BB provided the health economic analysis plan. AHY, MA and RZ provided oversight on the study procedures and delivery.

**Funding** The research has received funding from the National Institute for Health Research (NIHR) research for patient benefit (RfPB) (grant reference: PB-PG-0416-20039), the National Institute for Health Research (NIHR) Biomedical Research Centre at South London and Maudsley NHS Foundation Trust and King's College London and the Medical Research Council Doctoral Training Partnership (grant reference: 2064430).

**Competing interests** AHY is employed by King's College London as an honorary consultant in the South London and Maudsley Trust (NHS UK) and is a consultant to Johnson & Johnson and Livanova. AHY has given paid lectures and sat on advisory

boards for the following companies with drugs used in affective and related disorders: Astrazenaca, Eli Lilly, Lundbeck, Sunovion, Servier, Livanova, Janssen, Allegan, Bionomics, Sumitomo Dainippon Pharma. AHY has received honoraria for attending advisory boards and presenting talks at meetings organised by LivaNova. AHY is the Principal Investigator of the following studies: Restore-Life VNS registry study funded by LivaNova, ESKETINTRD3004: 'An Open-label, Long-term, Safety and Efficacy Study of Intranasal Esketamine in Treatment-resistant Depression', 'The Effects of Psilocybin on Cognitive Function in Healthy Participants'and 'The Safety and Efficacy of Psilocybin in Participants with Treatment-Resistant Depression (P-TRD)'. AHY has received grant funding (past and present) from the following: NIMH (USA); CIHR (Canada); NARSAD (USA); Stanley Medical Research Institute (USA); MRC (UK); Wellcome Trust (UK); Royal College of Physicians (Edin); BMA (UK); UBC-VGH Foundation (Canada); WEDC (Canada); CCS Depression Research Fund (Canada); MSFHR (Canada); NIHR (UK); Janssen (UK). AHY has no shareholdings in pharmaceutical companies. RZ is a private psychiatrist service provider at The London Depression Institute and co-investigator on a Livanova-funded observational study of Vagus Nerve Stimulation for Depression. RZ has received honorarium for talks at three medical symposia sponsored by Lundbeck as well as a talk funded by Janssen whom he collaborates with on an MRC-funded grant. KG reports grants from NIHR, Stroke association, National Institutes of Health (US) and Juvenile Diabetes Research Foundation (US) during the conduct of the study. EC reports personal fees from NIHR during the conduct of the study. BB reports grants from NIHR Research for Patient Benefit during the conduct of the study.

**Patient and public involvement** Patients and/or the public were involved in the design, or conduct, or reporting, or dissemination plans of this research. Refer to the Patient and public involvement section for further details.

**Patient consent for publication** Not required.

**Provenance and peer review** Not commissioned; externally peer reviewed.

**ORCID iDs**
Phillippa Harrison http://orcid.org/0000-0002-5039-7822
Mark Ashworth http://orcid.org/0000-0001-6514-9904

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
