## [Reviewer comments · BMJ Open]

ARTICLE DETAILS

TITLE (PROVISIONAL)	Study Protocol for The Antidepressant Advisor (ADeSS): A Decision Support System for Antidepressant Treatment for Depression in UK primary care - a feasibility study
AUTHORS	Harrison, Phillipa; Carr, Ewan; Goldsmith, Kimberley; Young, Allan; Ashworth, Mark; Fennema, Diede; Barrett, Barbara; Zahn, Roland

VERSION 1 – REVIEW

REVIEWER	Chad Bousman University of Calgary, Canada
REVIEW RETURNED	08-Dec-2019

GENERAL COMMENTS	The protocol is well written, the objectives are clear and the trial design is sound. However, I'm concerned with the design of the decision support tool. Although it is based on NICE guidelines, the tool fails to account for the wealth of pharmacokinetic evidence for the medications include in the algorithm. Escitalopram, citalopram and sertraline have CYP2C19-based dosing guidelines developed by the Clinical Pharmacogenetics Implementation Consortium (CPIC). An inability to tolerate or respond to one of these medications could indicate a metabolism issue (i.e., CYP2C19 poor metabolizers are less likely to tolerate these drugs and CYP2C19 ultrarapid metabolizers may not achieve the required plasma levels for efficacy). Likewise, vortioxetine, mirtazapine, venlafaxine, and paroxetine are CYP2D6 substrates of which, the latter two have CYP2D6-based prescribing guidelines. Thus, the algorithm in its current form will in some cases double or triple down on the same pharmacokinetic pathway. The algorithm would be strengthened by taking into account the pharmacokinetic pathway of each of the medications that an individual has not tolerated or responded to previously. By including this flexibility within the algorithm, there is a reduced chance of hitting the same and potentially perturbed pharmacokinetic pathway, which would in turn improve the the tool's utility. That all said, I'm sensitive to the fact that my suggested changes to the algorithm might not be feasible at this stage. Thus, at the minimum I would suggest the authors acknowledge and discuss this limitation to their decision support tool within the protocol. I would also suggest that Figure 1 be revised for better clarity and it should be included in the main text as it is the focus of the intervention under study.
--

REVIEWER	Ron Kessler Harvard Medical School, Department of Health Care Policy
REVIEW RETURNED	28-Dec-2019

GENERAL COMMENTS	I appreciate the desire to develop the infrastructure for implementing a decision support tool before the tool exists, but there are problems with this approach:  1. The optimal tool might require patients to fill out q'aires or submit to various biomarker assessments that will be critical factors in participation that cannot we studied here. 2. Clinician satisfaction, one of the outcomes here, would be importantly affected by the above requirements and by the success of the procedure, neither of which can be evaluated in the current study. I'm not sure an assessment of clinician satisfaction makes any sense in light of this fact. 3. The same problems arise in thinking about other outcomes:  a. Loss to follow-up will presumably be affected by burden of baseline assessment and effectiveness b. GP and px adherence similarly c. And the other outcomes of interest -- % of GPs willing to participate and recruitment rates per GP – strike me as interesting but hardly the sort of outcomes that would justify a registered trial rather than a small pilot study. Given the above, I'm not very positive about this study. In fact, I worry that it could lead to erroneous conclusions about the experiences that will occur in any subsequent trial when/if a popotentially useful decision support system is developed and evaluated.
---

REVIEWER	Peter Lucassen Radboud University Nijmegen, the Netherlands, Department of Primary and Community Care
REVIEW RETURNED	21-Jan-2020

GENERAL COMMENTS	Thank you for the opportunity to review this study protocol. This protocol is about a feasibility cluster-randomized trial of the effectiveness of a decision tool incorporated in the GPs' electronic medical system versus usual care. The objectives of the study are to develop a support tool for antidepressant treatment in primary care, to assess the feasibility of a future trial including estimates for computing effect size estimates, and to collect health economics data. Concerning the feasibility of the future trial the aim is to assess loss to follow up, GP and patient adherence, the number of GPs willing to participate, recruitment rates of patients and GP satisfaction with the tool. My comments:  1. The description of inclusion of participants is unclear to me. Partly, this is due to text dealing with this subject spread over different paragraphs on different pages. When I understand the authors well, inclusion starts with the use of the software eligibility tool (or EMIS search tool; is this the same instrument?). So, it is a kind of screening performed by the practice itself or by the research network. The results of this screening are provided to the GPs and the patients. The next step is asking consent and clinical evaluation of the patients by the researchers. The results of the assessment will be shared with GPs and patients. The patient will be told to consult their GP. The GP will use the Decision support tool and decide whether the patients needs antidepressant medication. Please provide a complete description of this process
---

	in one paragraph in the main text, not partly in supplementary files. Please correct my possibly false assumptions. It took me some time to reconstruct the process and I'm not sure about the correctness of my description. 2. Measurements in patients are partly enabled by the mobile phone app (MM-PHQ-9, Hypomanic symptoms (which questionnaire will be used?), FIBSER, medication changes and self-blamed action tendencies (which questionnaire will be used?); patients have to fill in a lot of other questionnaires, which are described in 'Data collection methods'. It would improve readability of the protocol when the authors would use one data collection paragraph. 3. The paragraph on Outcomes is not consistent (p 7). I would prefer to describe here the outcomes about feasibility; now it is a mix of outcomes of feasibility and trial outcomes. Please separate these issues. 4. I would add to the feasibility items: a. feasibility of the screening procedure, b. feasibility of using the mobile phone app by patients, c. feasibility of filling in the questionnaires by patients, d. reasons for GPs not to participate in the trial. Certainly for a. and d. I'm worried because the authors of this trial have – as far as I can see in their affiliations – no relation with primary care; I wonder if general practitioners have been involved in the development of this trial; this could be a major advantage in a priori solving some problems. 5. Although the authors have used the SPIRIT checklist, I think that the structure of the manuscript needs some reconsideration. It took me quite some time to discover the actual flow of the participants. For example: the paragraph 'Trial design' should be in the Methods section; the paragraph 'Adherence' should be part of the outcomes of the feasibility; 'Participant timeline' should be included in a paragraph 'Participants', the same goes for 'Eligibility criteria for patients'. In conclusion, the manuscript contains everything that is necessary for the description of a feasibility trial, but it is too difficult to read.
--	---

VERSION 1 – AUTHOR RESPONSE

Reviewer 1

Reviewer Name: Chad Bousman

Institution and Country: University of Calgary, Canada

Competing interests: None Declared

The protocol is well written, the objectives are clear and the trial design is sound. However, I'm concerned with the design of the decision support tool. Although it is based on NICE guidelines, the tool fails to account for the wealth of pharmacokinetic evidence for the medications include in the algorithm. Escitalopram, citalopram and sertraline have CYP2C19-based dosing guidelines developed by the Clinical Pharmacogenetics Implementation Consortium (CPIC). An inability to tolerate or respond to one of these medications could indicate a metabolism issue (i.e., CYP2C19 poor metabolizers are less likely to tolerate these drugs and CYP2C19 ultrarapid metabolizers may not achieve the required plasma levels for efficacy). Likewise, vortioxetine, mirtazapine, venlafaxine, and paroxetine are CYP2D6 substrates of which, the latter two have CYP2D6-based prescribing

guidelines. Thus, the algorithm in its current form will in some cases double or triple down on the same pharmacokinetic pathway. The algorithm would be strengthened by taking into account the pharmacokinetic pathway of each of the medications that an individual has not tolerated or responded to previously. By including this flexibility within the algorithm, there is a reduced chance of hitting the same and potentially perturbed pharmacokinetic pathway, which would in turn improve the the tool's utility. That all said, I'm sensitive to the fact that my suggested changes to the algorithm might not be feasible at this stage. Thus, at the minimum I would suggest the authors acknowledge and discuss this limitation to their decision support tool within the protocol.

Response: Thank you for these very thoughtful suggestions. We have incorporated these into the limitations section on page 18:

“One limitation of the algorithm is that it does not take pharmacogenomic evidence into account which could be an important refinement in the future, should pharmacogenomic testing become implemented into NHS routine care. For example, both CYP2D6 and CYP2C19 gene polymorphisms affect the liver metabolism of sertraline, escitalopram and citalopram, as well as paroxetine and fluvoxamine [43]. Therefore one could argue if someone has not responded to or not tolerated citalopram or paroxetine, this carries a higher risk of non-tolerance or non-response of escitalopram and sertraline. From a pragmatic perspective, however, it is worth trying escitalopram or sertraline as the first step in the algorithm, because pharmacogenomic effects of liver enzymes are only one of many factors which determine response and tolerance [44]. CYP2D6 also plays a role in metabolising vortioxetine and mirtazapine, but pharmacogenomic guidelines have not been developed for these to our knowledge.”

I would also suggest that Figure 1 be revised for better clarity and it should be included in the main text as it is the focus of the intervention under study.

Response: Thank you for this suggestion. We have incorporated a higher quality version of Figure 1 into the main manuscript (page 9). We would be happy to revise the content of Figure 1 if the reviewer could please guide us how to best revise it.

Reviewer 2

Reviewer Name: Ron Kessler

Institution and Country: Harvard Medical School, USA

Competing interests: None declared

I appreciate the desire to develop the infrastructure for implementing a decision support tool before the tool exists, but there are problems with this approach:

Response: We apologise for the confusion caused by including the goal of 'developing the decision support tool'; this was included as it was the first milestone of our grant application. However, development of the tool was not part of the feasibility trial, and in fact, this goal had been achieved prior to commencing the trial. Therefore, in the revised manuscript we have removed this goal (page 6) and added detail about the decision support tool to the feasibility aim (page 6):

“To assess the feasibility of a future confirmatory trial investigating the first computerised decision support tool for antidepressant treatment in UK primary care”.

We hope that this addresses the reviewer's concern.

1. The optimal tool might require patients to fill out q'aires or submit to various biomarker assessments that will be critical factors in participation that cannot be studied here.

Response: We do indeed gather information from patients via a mobile app which we have developed and this feeds into the decision support tool. While this could be enhanced by biomarkers, to date there is no licensed biomarker which accurately predicts treatment response and any such tool would need to demonstrate cost-effectiveness before being adopted by the NHS. Therefore, biomarkers as predictors of response are indeed an interesting area for further research, however this was outside of the remit of the designed intervention and feasibility trial.

2. Clinician satisfaction, one of the outcomes here, would be importantly affected by the above requirements and by the success of the procedure, neither of which can be evaluated in the current study. I'm not sure an assessment of clinician satisfaction makes any sense in light of this fact.
3. The same problems arise in thinking about other outcomes:
 - a. Loss to follow-up will presumably be affected by burden of baseline assessment and effectiveness
 - b. GP and px adherence similarly
 - c. And the other outcomes of interest -- % of GPs willing to participate and recruitment rates per GP – strike me as interesting but hardly the sort of outcomes that would justify a registered trial rather than a small pilot study.

Given the above, I'm not very positive about this study. In fact, I worry that it could lead to erroneous conclusions about the experiences that will occur in any subsequent trial when/if a potentially useful decision support system is developed and evaluated.

Response: Thank you for raising these points. This feasibility trial is designed to assess the feasibility of running a larger, confirmatory trial in the future. Many of the chosen outcomes are specifically intended to assess feasibility, rather than efficacy or cost-effectiveness. GP recruitment, adherence and satisfaction would be critical to the feasibility of a larger trial, and as such, are important to include here. Similarly, we agree that loss to follow-up is likely to be affected by the burden of baseline assessments, which is precisely why it is important to assess loss to follow-up in this feasibility trial. If GPs find the tool difficult to use and not helpful in making decisions, it would be inadvisable to proceed to a larger confirmatory trial and GPs will be able to evaluate whether the tool has been helpful for the patients they see in this feasibility trial.

Reviewer 3

Reviewer Name: Peter Lucassen

Institution and Country: Radboud University Nijmegen, the Netherlands,
Department of Primary and Community Care

Competing interests: None declared

Thank you for the opportunity to review this study protocol. This protocol is about a feasibility cluster-randomized trial of the effectiveness of a decision tool incorporated in the GPs' electronic medical system versus usual care. The objectives of the study are to develop a support tool for antidepressant treatment in primary care, to assess the feasibility of a future trial including estimates for computing effect size estimates, and to collect health economics data. Concerning the feasibility of the future trial the aim is to assess loss to follow up, GP and patient adherence, the number of GPs willing to participate, recruitment rates of patients and GP satisfaction with the tool.

1. The description of inclusion of participants is unclear to me. Partly, this is due to text dealing with this subject spread over different paragraphs on different pages. When I understand the authors well, inclusion starts with the use of the software eligibility tool (or EMIS search tool; is this the same instrument?).

Response: Thank you for highlighting this inconsistency. These two phrases "software eligibility tool" and "EMIS search tool" do indeed refer to the same instrument. As suggested, in the revised manuscript we use the term "EMIS search tool" throughout for consistency.

So, it is a kind of screening performed by the practice itself or by the research network. The results of this screening are provided to the GPs and the patients. The next step is asking consent and clinical evaluation of the patients by the researchers. The results of the assessment will be shared with GPs and patients. The patient will be told to consult their GP. The GP will use the Decision support tool and decide whether the patient needs antidepressant medication. Please provide a complete description of this process in one paragraph in the main text, not partly in supplementary files. Please correct my possibly false assumptions. It took me some time to reconstruct the process and I'm not sure about the correctness of my description.

Response: We apologise for the scattered presentation of this information, and agree this could be made much clearer. Your summary is entirely correct. In the revised manuscript, we have rewritten the “Participant Timeline” on page 8 to clarify screening procedures:

“Potential participants will be identified using an EMIS eligibility tool to search participating GP practice medical records for patients meeting inclusion criteria (see Supplementary material). Interested participants will complete a pre-screening assessment, followed by a screening assessment including a clinical evaluation to determine full eligibility. The results of the screening assessment will be shared with GPs and patients. Inclusion decisions will not be made by GPs to help mitigate against selection biases and will be decided by the research team after the screening assessment. After being enrolled, patients receive access to a secure mobile phone app developed in the study. Eligible patients will undergo treatment for depression over 14 weeks with their GPs. In both arms, patients will be seen by their GP who will review and monitor treatment. Patients’ participation will last for approximately 15-18 weeks after their face-to-face assessment, at which point their follow-up assessment will be conducted. See Table 1 below for participant timeline.”

2. Measurements in patients are partly enabled by the mobile phone app (MM-PHQ-9, Hypomanic symptoms (which questionnaire will be used?), FIBSER, medication changes and self-blamed action tendencies (which questionnaire will be used?); patients have to fill in a lot of other questionnaires, which are described in ‘Data collection methods’. It would improve readability of the protocol when the authors would use one data collection paragraph.

Response: Thank you for highlighting the lack of clarity here. The feasibility trial mobile app includes two screening questions from the WHO CIDI self-rated version with kind permission of the author (Kessler et al., 2006) to assess hypomanic symptoms (a reference for this is provided on page 15). It also includes two self-blame-related questions developed in Dr Zahn’s lab, described on page 15. In the revised manuscript we have now made the following changes:

□ To make clear that all measures described under “exploratory clinical outcomes” are self-reported via the mobile app, we have moved “self-report via mobile app” to under the “Outcomes” section to the subtitle “Exploratory clinical outcome measures (self-report via mobile app)” on page 11.

□ We have moved the description of the app measures to “Data collection methods” on page 15 to provide a more coherent description of all measures in one place:

“Participants will use the mobile app for the 14-week study duration to enter weekly MM-PHQ-9 ratings [18], hypomanic symptoms [24], FIBSER medication side effects [19], medication changes and self-blame-related action tendencies such as to what extent participants would feel like hiding or creating a distance from themselves (using two questions developed in Dr Zahn’s research lab). A daily question will ask about medication adherence. Alternatively, participants can opt to provide this data at weekly intervals via post or phone.”

(“Frequency, Intensity and Burden of Side Effects Ratings” has been described in full on page 11 due to its deletion from page 6).

(“alternatively over the telephone or via postal survey” has been added to details of the pre-screening under “Data Collection Methods” on page 15 to clarify that alternative methods to online are available.

3. The paragraph on Outcomes is not consistent (p 7). I would prefer to describe here the outcomes about feasibility; now it is a mix of outcomes of feasibility and trial outcomes. Please separate these issues.

Response: Thank you for raising this issue; we agree this section could be clearer. We had intense discussions with our trial statisticians about how best to present the outcomes. In the revised manuscript, we have hopefully improved the clarity of this section by labelling the clinical outcome measures for a future trial as

“This section describes outcomes that will be measured in this feasibility trial to provide estimates (e.g. effect size, standard deviation) needed for sample size calculations for a future confirmatory trial. These outcomes will not be used for inferential comparisons between trial arms.”

“Primary/secondary clinical outcomes for a future confirmatory trial:” (p. 11). A description of this section has been included on page 11:

This hopefully makes it clear which outcomes are feasibility trial outcome measures and which are for the purpose of informing a future confirmatory trial.

4. I would add to the feasibility items: a. feasibility of the screening procedure, b. feasibility of using the mobile phone app by patients, c. feasibility of filling in the questionnaires by patients, d. reasons for GPs not to participate in the trial. Certainly for a. and d. I'm worried because the authors of this trial have – as far as I can see in their affiliations – no relation with primary care; I wonder if general practitioners have been involved in the development of this trial; this could be a major advantage in a priori solving some problems.

Response: We agree that it is important to assess feasibility of screening procedures and the mobile phone app. We carefully selected these feasibility outcomes through collaboration with primary care experts and statisticians prior to the trial commencing in August 2018, which have been approved by our ethics committee and registered on clinicaltrials.gov. We plan to formally measure feasibility of using the mobile phone app with a dedicated questionnaire for patients about their usage and satisfaction with the app and any difficulties they faced. Additionally, we will assess GP satisfaction with the advisor tool and the mobile app using a dedicated questionnaire. We believe that these are key feasibility outcomes because they assess patient and GP satisfaction with the processes that would be replicated in a future confirmatory trial. Further assessments will be made informally through interactions with patients and GPs and reported as exploratory feasibility outcomes in the final report, including collecting reasons for withdrawal of GPs and feedback on the screening procedures from patients. These outcomes will be very informative for the design of a future confirmatory trial.

Regarding the primary care expertise which underpins the design of our trial and instruments, we have a very strong team with our co-investigator Dr Mark Ashworth, who is an academic GP with experience as a medical director of a large GP surgery in south London, as well as Dr Daniel Dietch who is a clinical GP and who has advised us extensively in his role as chair of our steering committee. We should also mention that the chief investigator has run a primary care consultation liaison service for mood disorders in South London which allowed him to gather important insights into GPs' views and use of electronic health records as well as prescribing practices. Last but not least, we should mention that our subcontractor EMIS PLC is run by a practicing GP and that our tool has undergone initial quality control and user testing by GPs associated with EMIS.

5. Although the authors have used the SPIRIT checklist, I think that the structure of the manuscript needs some reconsideration. It took me quite some time to discover the actual flow of the participants. For example: the paragraph 'Trial design' should be in the Methods section; the paragraph 'Adherence' should be part of the outcomes of the feasibility; 'Participant timeline' should be included in a paragraph 'Participants', the same goes for 'Eligibility criteria for patients'.

Response: Thank you for pointing out the confusing structure. We have taken this on board and, in the revised manuscript, restructured the manuscript such that “Trial design” is described in the Methods section (page 7); a “Participants” section has been added (page 7) comprising “Eligibility criteria for patients” (page 7) and “Participant Timeline” (page 8). We have kept the “Adherence” paragraph as a separate section to the feasibility outcomes, following SPIRIT guidelines, to describe patient and GP adherence to study procedures in more detail. However, we have also listed adherence under the feasibility outcomes, as you suggest, to emphasise the importance of this feasibility outcome.

In conclusion, the manuscript contains everything that is necessary for the description of a feasibility trial, but it is too difficult to read.

Response: Thank you for these helpful comments. We hope that our changes have improved the clarity of the planned feasibility trial, and that the manuscript is now easier to read.

VERSION 2 – REVIEW

REVIEWER	Chad Bousman University of Calgary
REVIEW RETURNED	09-Mar-2020

GENERAL COMMENTS	The authors have addressed my concerns. However, the the following sentence should be revised: "From a pragmatic perspective, however, it is worth trying escitalopram or sertraline as the first step in the algorithm, because pharmacogenomic effects of liver enzymes are only one of many factors which determine response and tolerance [44]." Citation #44 does not support the use of escitalopram or sertraline as a first step in treatment. However, citation #44 does state that there are many factors which determine response and tolerance to medications. These are two separate statements and only the latter should be linked to citation #44.
--

REVIEWER	Peter Lucassen Radboud University Nijmegen Department of Primary and Community Care The Netherlands
REVIEW RETURNED	05-Mar-2020

GENERAL COMMENTS	Thank you for addressing the points that I have raised in the previous version. I consider the description of the trial to be complete and according to current standards. The manuscript reads much easier now.
--

VERSION 2 – AUTHOR RESPONSE

Reviewer 1

Reviewer Name: Chad Bousman

Institution and Country: University of Calgary, Canada

Competing interests: None Declared

The authors have addressed my concerns. However, the the following sentence should be revised: "From a pragmatic perspective, however, it is worth trying escitalopram or sertraline as the first step in the algorithm, because pharmacogenomic effects of liver enzymes are only one of many factors which determine response and tolerance [44]." Citation #44 does not support the use of escitalopram or sertraline as a first step in treatment. However, citation #44 does state that there are many factors which determine response and tolerance to medications. These are two separate statements and only the latter should be linked to citation #44.

Response: Thank you for making us aware of this. We have now separated the sentences in the revised manuscript (pg 16) so that citation #44 is only linked to the latter statement, as below:

“Pharmacogenomic effects of liver enzymes, however, are only one of many factors which determine response and tolerance [44]. From a pragmatic perspective it is therefore worth trying escitalopram or sertraline as the first step in the algorithm which is supported by network-meta-analytical evidence [45].